# Drp1 Overexpression Decreases Insulin Content in Pancreatic MIN6 Cells

**DOI:** 10.3390/ijms232012338

**Published:** 2022-10-15

**Authors:** Uma D. Kabra, Noah Moruzzi, Per-Olof Berggren, Martin Jastroch

**Affiliations:** 1Division of Metabolic Diseases, Technische Universität München, 80333 Munich, Germany; 2Department of Pharmaceutical Chemistry, Parul Institute of Pharmacy, Parul University, Vadodara 391760, Gujarat, India; 3The Rolf Luft Research Center for Diabetes and Endocrinology, Karolinska Institutet, Karolinska University Hospital, SE-171 76 Stockholm, Sweden; 4Department of Molecular Biosciences, The Wenner-Gren Institute, Stockholm University, SE-106 91 Stockholm, Sweden

**Keywords:** dynamin-related protein 1, glucose-stimulated insulin secretion, insulin content, overexpression, MIN6 cell, bioenergetics

## Abstract

Mitochondrial dynamics and bioenergetics are central to glucose-stimulated insulin secretion by pancreatic beta cells. Previously, we demonstrated that a disturbance in glucose-invoked fission impairs insulin secretion by compromising glucose catabolism. Here, we investigated whether the overexpression of mitochondrial fission regulator *Drp1* in MIN6 cells can improve or rescue insulin secretion. Although *Drp1* overexpression slightly improves the triggering mechanism of insulin secretion of the *Drp1*-knockdown cells and has no adverse effects on mitochondrial metabolism in wildtype MIN6 cells, the constitutive presence of *Drp1* unexpectedly impairs insulin content, which leads to a reduction in the absolute values of secreted insulin. Coherent with previous studies in *Drp1*-overexpressing muscle cells, we found that the upregulation of ER stress-related genes (*BiP*, *Chop*, and *Hsp60*) possibly impacts insulin production in MIN6 cells. Collectively, we confirm the important role of *Drp1* for the energy-coupling of insulin secretion but unravel off-targets effects by *Drp1* overexpression on insulin content that warrant caution when manipulating *Drp1* in disease therapy.

## 1. Introduction

Mitochondria play a central role in cellular bioenergetics, specifically in pancreatic beta (*β*) cells by generating chemical energy in the form of ATP through oxidative phosphorylation [1]. Alongside this, mitochondria are also engaged in the dynamic activity of fusion and fission to maintain healthy mitochondria. The molecular machinery that regulates these dynamic processes belongs to the family of large GTPases. Mitofusins 1/2 (*Mfn 1/2*) and Optic atropy 1 (*Opa 1*) are involved in mitochondrial fusion, whereas mitochondrial fission is mainly carried out by Dynamin-related protein 1 (*Drp1*) [2]. Although it is well known that mitochondrial dynamics and metabolism are interrelated, the impact of mitochondrial dynamics on bioenergetics has been overlooked [3]. The data unfolds the relation between mitochondrial dynamics and metabolism, well observed in islets from type 2 diabetic subjects that showed impaired insulin secretion, which has been attributed to altered mitochondrial morphology, reduced glucose oxidation, and ATP production [4]. In light of this, several studies support the notation that overexpression or downregulation of mitochondrial proteins can alter mitochondrial structure and consequently glucose-stimulated insulin secretion (GSIS) in *β* cells. For instance, the deletion of mouse *Opa1* from *β* cells leads to impaired glucose-stimulated ATP production, making mice hyperglycemic [5,6]. Furthermore, *Mfn1* overexpression in INS-1E cells caused dramatic mitochondrial aggregation and, as a consequence, impaired insulin secretion [7]. Similarly, overexpression of dominant-negative *Drp1* (*DN-Drp1*) in INS-1E cells reduced insulin secretion due to increased mitochondrial proton leak [8]. Previously, we demonstrated that genetic or pharmacologic silencing of *Drp1* impaired GSIS in MIN6 cells and islets due to decreasing glucose-fueled respiration, rather than due to mitochondrial proton leak as reported earlier [9]. Several results demonstrate that the balance between the mitochondrial fission/fusion machinery is a prerequisite for appropriate insulin secretion in *β* cells.

In this paper, we transiently overexpressed *Drp1* to study whether it can rescue the impaired insulin secretion in *Drp1*-knockdown (*Drp1*-KD) MIN6 cells. We discovered that *Drp1* overexpression in *Drp1*-KD cells only slightly improved insulin secretion but failed to rescue insulin secretion to wildtype levels. To explore the molecular underpinnings, we overexpressed *Drp1* in wildtype MIN6 cells and found that *Drp1* overexpression did not alter the concentration of mitochondrial respiratory complexes, consistent with unchanged glucose-stimulated respiration and ATP content. However, we discovered decreased insulin content in *Drp1*-overexpressing cells. Importantly, appropriate normalization of glucose-triggered insulin secretion to insulin content, as suggested previously [10], reveals that the bioenergetic triggering mechanism for insulin release is rather improved in *Drp1*-overexpressing cells. Thus, the underlying defect for reduced insulin secretion can be attributed to reduced insulin content. We suggest that constitutively elevating *Drp1* impairs insulin biosynthesis in MIN6 cells, which is corroborated by the activation of the *PKA/eIF2α/Fgf21* pathway, indicating ER stress and reduction of protein translation as shown previously in muscle cells [11]. Collectively, our data suggest that, despite the important role of *Drp1* during GSIS, the overexpression of *Drp1* is not instrumental for future therapy due to off-target effects on insulin biosynthesis.

## 2. Results

### 2.1. Drp1 Overexpression Does Not Rescue GSIS in Beta Cells

*Drp1* was transiently overexpressed in *Drp1*-KD MIN6 cells with 2.5 and 5 µg of plasmid DNA using the electroporation technique. Overexpression of *Drp1* in *Drp1*-KD cells with 2.5 µg of DNA resulted in the increase of *Drp1* level to wildtype cells, whereas with 5 µg of DNA, supraphysiological levels were reached (Figure 1A). *Drp1* overexpression significantly lowers basal insulin secretion and also fails to restore the impaired GSIS in *Drp1*-KD cells (Figure 1B).

### 2.2. Drp1 Overexpression Altered Mitochondrial Morphology But Had No Effect on Mitochondrial Respiratory Complexes in MIN6 Cells

To mechanistically understand the failure of rescuing insulin secretion, we transiently overexpressed *Drp1* in wildtype MIN6 cells, as the knockdown may have imposed secondary effects. This overexpression protocol resulted in a ~5-fold increase in *Drp1* mRNA levels (Figure 2A) and a ~3-fold increase in *Drp1* protein levels (Figure 2B) compared to the control cells. *Drp1* overexpression did not affect *Mfn1* or *Opa1* mRNA levels but caused a slight decrease in *Mfn2* mRNA, in agreement with the notion that mitochondrial fragmentation is increased (Figure 2C). Furthermore, fluorescent visualization of mitochondria suggested an increased abundance of fragmented mitochondria in *Drp1*-overexpressing cells compared to control cells (Figure 2D). *Drp1* overexpression does not affect components of the mitochondrial electron transport chain as confirmed by Western blot analysis of respiratory complex subunits (Figure 2E and Appendix A).

### 2.3. Reduced Insulin Content in Drp1-Overexpressing MIN6 Cells

*Drp1* overexpression caused a 30% reduction of insulin content in MIN6 cells (Figure 3A). Unspecific off-target effects of protein overexpression per se could be excluded, as no reduction of insulin was seen when expressing a random protein, such as GFP (Appendix A). Additionally, *Ins1* mRNA levels were also decreased, while no changes in *Ins2* mRNA levels were observed (Figure 3B). Upon *Drp1* overexpression in the wildtype cells, we found a significant decrease of secreted insulin per microgram of DNA (Figure 3C), identical to the rescue attempts in *Drp1*-KD cells. Differences in insulin content mask bioenergetic deficits of insulin triggering [10]. Therefore, we normalized the secreted insulin values to the insulin content. This eliminated differences in basal insulin secretion at 2 mM glucose and revealed significant triggering of insulin release at 16.5 mM glucose (Figure 3D), demonstrating improved energy-coupled insulin secretion in *Drp1*-overexpressing cells. We measured the oxygen consumption of the cells using plate-based respirometry and depicted the aggregated oxygen consumption rates for the entire course of the assay (Figure 3E). A comprehensive analysis of mitochondrial bioenergetic parameters revealed no differences in glucose-stimulated, proton leak, and ATP-linked respiration, as well as coupling efficiency, between control and *Drp1*-overexpressing cells (Figure 3F–I). In agreement with the respirometric parameters, *Drp1* overexpression did not affect intracellular ATP content before and after glucose activation (Figure 3J).

### 2.4. Stress Pathway Activation upon Drp1-Overexpression in MIN6 Cells

In order to investigate potential mechanisms responsible for the reduction of insulin content, we studied the expression levels of stress-related pathways, as misfolding of secretory proteins can lead to ER stress and increase the UPR, possibly causing a reduction in insulin transcription, translation, and promoting apoptosis. We found increased expression of ER and UPR stress-related genes (Figure 4A–C) in *Drp1*-overexpressing MIN6 cells, suggesting the involvement of ER stress in the decreased production of insulin in MIN6 cells. Based on our findings, we summarized the impact of *Drp1* in pancreatic *β* cells (Figure 4D).

## 3. Discussion

The mitochondrial fission protein *Drp1* is an important regulator of insulin secretion as it takes part in the machinery that fragments mitochondria upon glucose stimulation [4,5,7,8,9]. In this study, we demonstrate that *Drp1* overexpression in MIN6 cells is not instrumental to rescue or improve glucose-stimulated insulin secretion. This is due to a significant reduction in insulin content, which is presumably caused by induction of the *PKR/eIF2α/Fgf21* pathway, most likely hampering insulin biosynthesis, coherent with previous observations of *Drp1* overexpression in muscle cells [11].

In pancreatic *β* cells, alteration in mitochondrial fission/fusion proteins has been shown to affect mitochondrial morphology leading to type 2 diabetes (T2D) [12]. The expression of *Drp1* was found to be increased in the islets of a type 2 diabetis mouse model [13,14,15]. Accumulating evidence has confirmed that tight regulation of mitochondrial fission by *Drp1* is important for appropriate GSIS [13,16]. In line with this, in our previous work, we showed that genetic or pharmacologic inhibition of *Drp1* impairs mitochondrial bioenergetics and alters GSIS in *β* cells and islets [9]. In the present study, we overexpressed *Drp1* in *Drp1*-KD MIN6 cells to rescue the impaired insulin secretion. Even after attaining expression of *Drp1* to comparable levels of endogenous *Drp1*, decreased insulin secretion in *Drp1*-KD MIN6 cells was not rescued or improved. At this stage, it can only be speculated that the complex machinery of fission–fusion requires several dynamic adjustments to improve insulin secretion.

Overexpression of *Drp1* in wildtype MIN6 cells modulated mitochondrial morphology but had no demonstrable effect on the concentration of mitochondrial respirometry complex subunits, cellular respiration, and ATP content. This is an important observation as mitochondrial oxidative phosphorylation is crucial for the metabolism–secretion coupling of pancreatic *β* cells [17,18,19]. *Drp1* overexpression, however, caused a significant decrease in the absolute amount of secreted insulin. In our previous work, we revealed that secreted insulin values require normalization to insulin content to establish a robust linear relationship between insulin secretion and bioenergetic parameters [10]. Normalization of secreted insulin to insulin content revealed that *Drp1* overexpression even improved energy-coupling. Thus, the reduced insulin secretion can be fully attributed to reduced insulin content in the *Drp1*-overexpressing cells. The reduced insulin content is coherent with a slight reduction in *Ins1* gene expression, partially explaining the phenotype.

However, we propose that the reduced insulin content is mainly caused by problems of insulin biosynthesis and at the level of the endoplasmic reticulum (ER), which is an important organelle for protein biogenesis in *β* cells. Accumulating evidence revealed that any impairment, e.g., due to aggregation of misfolded protein, develops ER stress and dysfunction, which further contributes to the *β* cell failure [20,21]. ER stress triggers the unfolded protein response (UPR) through the activation of transmembrane-protein-like Inositol-requiring protein 1 (*Ire1*), PKR-like endoplasmic reticulum kinase (*Perk*), and Activating transcription factor (*Atf-6*), which in turn enhances protein folding capacity and inhibits protein synthesis. However, persistent activation of the UPR induces *β* cell apoptosis [22,23,24]. Previous studies showed that *β* cells are more prone to ER stress as they are involved in the synthesis, regulation, and secretion of insulin, for e.g., increased expression of UPR markers has been reported in an animal model and in humans with T2D [25,26,27,28].

In *β* cells, the mitochondrial fission protein *Drp1* resides on the ER [29]. Previous studies showed that stressing *β* cells with glucose or palmitate induces ER stress alongside increases of *Drp1* expression and apoptosis [30,31]. Hence, *Drp1* is presumably implicated in ER-stress-induced *β* cells apoptosis and reduced GSIS. We report that overexpression of wildtype *Drp1* activates the ER-stress-related pathway, which interferes with the production of insulin. This is coherent with observations by Touvier *et al*., reporting that overexpression of *Drp1* in skeletal muscle cells activates mitochondrial-stress-related genes and inhibits protein translation [11]. Collectively, our data reveal for the first time how disturbance in mitochondrial dynamics influences insulin synthesis in pancreatic *β* cells.

## 4. Materials and Methods

Mouse monoclonal anti-*Drp1* antibody (611113), total OXPHOS rodent WB antibody cocktail (ab110413), and mouse monoclonal anti-*α*-tubulin antibody (sc-23948) were purchased from BD Biosciences, Heidelberg, Germany; Abcam, Cambridge, MA, USA; and Santa Cruz Biotechnology. The mouse anti-rabbit IgG-HRP antibody (sc-2357) and rabbit anti-mouse IgG-HRP antibody (sc-358914) were obtained from Santa Cruz Biotechnology, CA, USA. All the primers for qPCR were purchased from Sigma-Aldrich, Taufkirchen, Germany. *Drp1* (FW: 5′-TAAGCCCTGAGCCAATCCATC-3′; RV:5′-CATTCCCGGTAAATCCACAAGT-3′), *Ins1* (FW: 5′- CACTTCCTACCCCTGCTGG-3′; RV:5′-ACCACAAAGATGCTGTTTGACA-3′), *Ins2* (FW: 5′- GCTTCTTCTACACACCCATGTC-3′; RV:5′-AGCACTGATCTACAATGCCAC-3′), *Mfn1* (FW: 5′- CCTACTGCTCCTTCTAACCCA-3′; RV:5′-AGGGACGCCAATCCTGTGA-3′), *Mfn2* (FW: 5′-ACCCCGTTACCACAGAAGAAC -3′; RV:5′- AAAGCCACTTTCATGTGCCTC -3′), *Opa1* (FW: 5′- TGGAAAATGGTTCGAGAGTCAG-3′; RV:5′- CATTCCGTCTCTAGGTTAAAGCG-3′), *BiP* (FW: 5′-TTCAGCCAATTATCAGCAAACTCT-3′; RV: 5′-TTTTCTGATGTATCCTCTTCACCAGT-3′), *Chop* (FW: 5′-CCACCACACCTGAAAGCA-3′; RV: 5′-AGGTGAAAGGCAGGGACTCA-3′), *Atf4* (FW: 5′-GGGTTCTGTCTTCCACTCCA-3′; RV: 5′-AAGCAGCAGAGTCAGGCTTTC), *Grp94* (FW:5′-AAGAATGAAGGAAAAACAGGACAAAA-3′; RV: 5′-CAAATGGAGAAGATTCCGCC-3′), *Fgf21* (FW: 5′- GCTGCTGGAGGACGGTTACA-3′; RV: 5′-CACAGGTCCCCAGGATATTG-3′), *Hsp60* (FW: 5′-GCAGAGTTCCTCAGAAGTTGG-3′; RV: 5′- GCATCCAGTAAGGCAGTTCTC-3′), *Atpif1* (FW: 5′- GGTGTCTGGGGTATGAAGGTC-3′; RV: 5′-CCTTTTCTCGTTTTCCGAAGGC-3′). All reagents used were of analytical grade and purchased from Sigma-Aldrich, Taufkirchen, Germany.

Cell culture: MIN6 mouse clonal *β* cells were provided by Prof. J. Miyazaki (Osaka University, Japan) and grown in Dulbecco’s modified eagle’s medium (DMEM) with glutaMAX containing 25 mM glucose (Thermo Fisher Scientific, Waltham, MA, USA) supplemented with 15% heat-inactivated Hyclone^TM^ Fetal Bovine Serum (Thermo Fisher Scientific, Waltham, MA, USA), 72 µM 2-mercaptoethanol, 100 µg/mL penicillin, and 100 µg/mL streptomycin. The cells were maintained in a humidifier (37 °C), containing 5% CO_2_ and 95% air environment.

Lentivirus infection: Lentiviral pLKO.1 control shRNA and mouse *Drp1* shRNA plasmid (Clone ID: NM_152816.1-1101s1c) were purchased from Sigma-Aldrich, Taufkirchen, Germany. The stable cell line was established by infecting MIN6 cells with the lentivirus (MOI = 1). Transduced MIN6 cells were cultured for 24 h and then selected with 1 µg/mL puromycin for several days until a stable, puromycin-resistant cell population was obtained.

Plasmid DNA extraction and transient transfection: The empty vector pcDNA3.1 (Plasmid #138209) and pcDNA3.1 (+) *Drp1* (the latter was gifted by David Chan plasmid # 34706) were purchased from Addgene. The Luria broth (LB) agar plates were prepared by mixing precooled LB media at 55 °C with ampicillin 100 μg/mL. Strains were streaked onto LB agar plates and spread evenly. The plates were then incubated for 16 h at 37 °C. The next day, a single colony was picked from the LB plates and suspended in 5 mL LB media containing ampicillin and incubated overnight at 37 °C with shaking. After overnight culturing, the bacterial culture was centrifuged, and DNA plasmid mini-isolation was performed using a QIAprep Spin Miniprep kit (Qiagen GmBH, Hilden, Germany) according to the manufacturer’s protocol. Plasmid DNA concentration was measured using Nanodrop 2000 UV–Visible spectrophotometer (Thermo Fisher Scientific, Waltham, MA, USA). The cells were transiently transfected with 5 µg of plasmid DNA/cuvette by electroporation using a Nucleofector kit V (Lonza, Cologne, Germany) as per the manufacturer’s protocol.

Insulin secretion from MIN6 cells: Briefly 30,000 cells/well were seeded into a 96-well plate containing DMEM containing 25 mM glucose. After 48 h, the medium was replaced with DMEM containing 5 mM glucose. Later after 16 h, the cells were starved for 2 h in Krebs–Ringer (KRH) buffer containing 2 mM glucose. The starvation medium was then replaced with KRH buffer supplemented with different glucose concentrations. After incubation for 2 h, the secreted insulin levels and total insulin content were determined using a mouse ultrasensitive insulin ELISA kit (Alpco, Salem, NH, USA). For normalization, the DNA content of the cells was measured using the Quant-iT™ PicoGreen™ dsDNA Assay kit (Invitrogen, Darmstadt, Germany).

ATP measurement: Around 10,000 cells/well were seeded into white 96-well plates and cultured in DMEM containing 25 mM glucose. The cells were treated similarly to the insulin secretion assay. At the end of the experiment, the medium was aspirated, and ATP amounts were determined using the Luminescent ATP detection assay kit (Abcam, Waltham, MA, USA) according to the manufacturer’s protocol. Results were corrected for DNA content.

Oxygen consumption: Measurement of oxygen consumption in cells was performed using a Seahorse XF24 analyzer according to the manufacturer’s instructions (Agilent, Seahorse Bioscience, Santa Clara, CA, USA) and practical guides [32]. Cells were seeded at a density of 40,000 cells/well into an XF24-well plate in DMEM containing 25 mM glucose. After 48 h, the medium was replaced with DMEM containing 5 mM glucose. Later after 16 h, the cells were starved for 2 h in bicarbonate-free Krebs–Ringer (KRH) buffer containing 2 mM glucose. The plate was then transferred into the machine, and after the completion of calibration, the program was started. After four measurement cycles of basal cellular respiration, cells were stimulated with glucose (16.5 mM, 20 cycles), followed by oligomycin (10 μg/mL, 3 cycles), and a mixture of rotenone/antimycin A (R/A; 1/2 μM, 3 cycles). Respiration values are corrected for non-mitochondrial respiration. For normalization, the DNA content of the cells was measured. The individual bioenergetics parameters of OXPHOS were calculated according to [9].

Western blotting: Cells were washed with cold Dulbecco’s phosphate-buffered saline (Thermo Fisher Scientific, Waltham, MA, USA) and incubated in cold RIPA lysis buffer (Thermo Fisher Scientific, Waltham, MA, USA) with a cocktail of protease and phosphatase inhibitors, followed by sonication for 10 s. Cell lysates were centrifuged at 12,000× *g* for 10 min at 4 °C, and protein concentration was quantified using the Pierce^TM^ BCA Protein Assay kit (Thermo Fisher Scientific, Waltham, MA, USA) according to the manufacturer’s protocol. An equal amount of protein lysates was separated by sodium dodecyl sulfate–polyacrylamide gel electrophoresis (SDS-PAGE) and transferred onto nitrocellulose membranes. The membranes were blocked with 5% BSA prepared in TBST for 1 h at room temperature and then probed overnight at 4 °C with the respective primary antibodies. After washing, the membranes were incubated with horseradish peroxidase (HRP)-conjugated secondary antibodies in TBST containing 5% BSA for 1 h at room temperature. Membranes were then developed using a chemiluminescence detection system (LI-COR Biosciences, Lincoln, NE, USA). The intensity of each band was quantified by densitometry using Image J software, 1.53t, NIH, USA.

RNA isolation and qPCR: Total RNA from the treated cells was extracted using a RNeasy Mini kit (Qiagen GmBH, Hilden, Germany) according to the manufacturer’s protocol. Quantification of all total RNA samples was performed using a NanoDrop 2000 UV–Visible spectrophotometer (Thermo Fisher Scientific, Waltham, MA, USA). Briefly, 1 µg of total RNA was used to synthesize cDNA using the QuantiTect Reverse Transcription kit (Qiagen GmBH, Hilden, Germany) according to the manufacturer’s protocol. qPCR was performed with SYBR-green using a ViiA 7 Real-Time PCR system (Applied Biosystem, Foster City, USA). The fold induction was calculated by the ΔΔCt method using *HPRT* as the control gene.

Statistical analysis: Statistical analysis was performed with GraphPad Prism version 9.0, San Diego, CA, USA. Data were collected from different independent experiments. Unpaired two-tailed Student’s *t* tests were used to compare two variables, and one-way ANOVA (with Bonferroni post hoc analysis) was used for multiple comparisons. All the data are shown as mean ± standard error of the mean (S.E.M). Statistically significant differences were considered at *p* < 0.05 (*), *p* < 0.01 (**), *p* < 0.001 (***), *p* < 0.0001 (****).

## Figures and Tables

**Figure 1 ijms-23-12338-f001:**
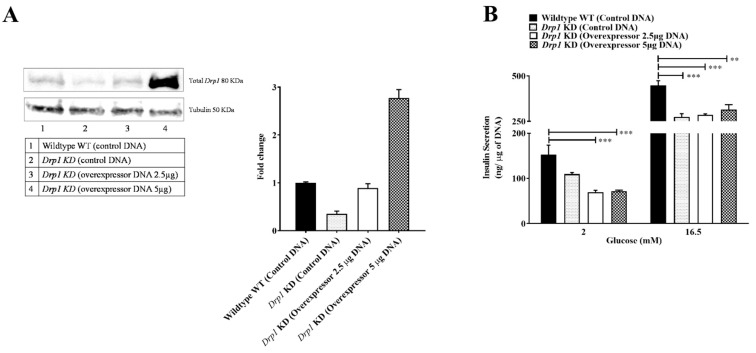
Insulin secretion of *Drp1*-KD MIN6 cells overexpressing *Drp1*. *Drp1*-KD MIN6 cells were transiently transfected with *Drp1*-containing plasmid using electroporation. (**A**) Immunoblot and densitometric quantification of overexpressed *Drp1*. Data are represented as mean ± SEM (*n* = 3). (**B**) Insulin release is expressed as nanograms of insulin per microgram of DNA. Data are presented as mean ± SEM (*n* = 4), and *n*-values represent independent experiments. The statistical significance of mean differences was tested by one-way ANOVA (with Bonferroni post hoc analysis) for multiple comparisons. *p* < 0.01 (**), *p* < 0.001 (***).

**Figure 2 ijms-23-12338-f002:**
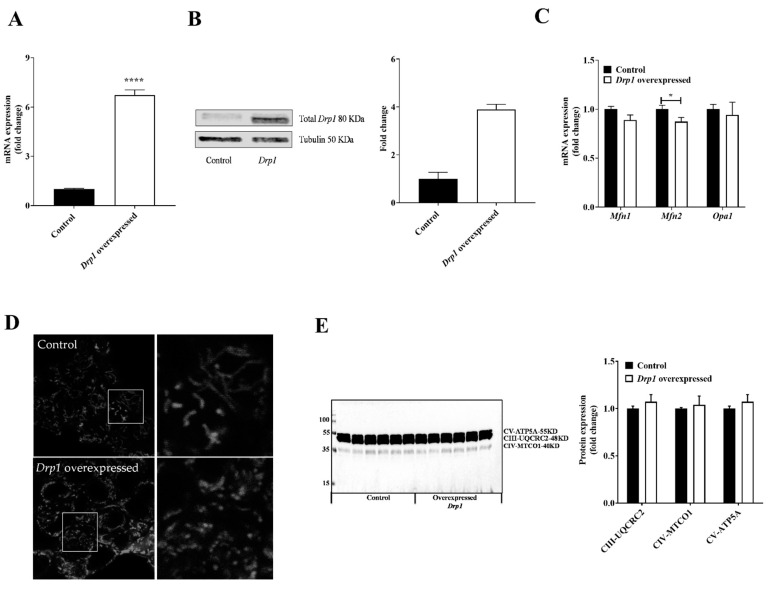
Alteration of mitochondrial morphology but not respiratory machinery by *Drp1* overexpression in MIN6 cells. MIN6 cells were transiently transfected using *Drp1* plasmid by the electroporation technique to overexpress *Drp1*: control (black bars) and overexpressed *Drp1* (white bars). (**A**) The overexpression efficiency of *Drp1* was confirmed by qPCR. *HPRT* was used as a control. (**B**) Representative immunoblot and densitometric quantification of *Drp1* protein content. Tubulin was used as a loading control. (**C**) Relative levels of *Mfn1*, *Mfn2,* and *Opa1* mRNA were measured by qPCR, using *HPRT* as the housekeeper gene. (**D**) Live confocal imaging of transiently transfected MIN6 cells that were stained with MitoTracker Red FM for 30 min. Representative confocal images of the control and *Drp1*-overexpressed cells. (**E**) Immunoblot and densitometric quantification of OXPHOS complexes. Data are represented as mean ± SEM (*n* = 6), and *n*-values represent independent experiments. The statistical significance of mean differences was tested by an unpaired two-tailed Student’s *t*-test. *p* < 0.05 (*), *p* < 0.0001 (****).

**Figure 3 ijms-23-12338-f003:**
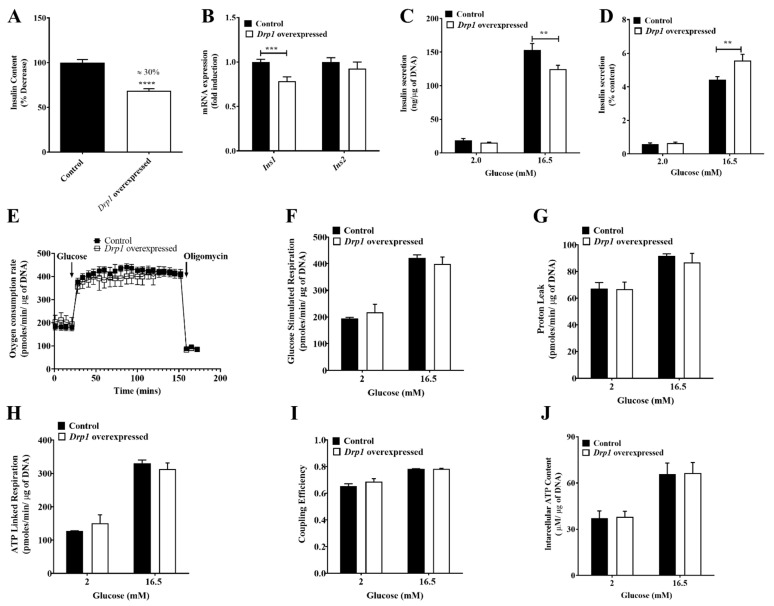
Insulin content and GSIS of *Drp1*-overexpressed MIN6 cells. MIN6 cells were transiently transfected with the *Drp1* plasmid by the electroporation technique to overexpress *Drp1;* control (black bars) and overexpressed *Drp1* (white bars). (**A**) Insulin content expressed as a % decrease. (**B**) Relative levels of *Ins1* and *Ins2* mRNA were measured by qPCR. *HPRT* was used as a control. Data are represented as mean ± SEM (*n* = 3). (**C**) Insulin release is expressed as nanograms per microgram of DNA. (**D**) Insulin release is expressed as a percentage of content. Data are represented as mean ± SEM (*n* = 4). (**E**) Averaged, time-resolved oxygen consumption traces measured with the XF24 extracellular flux analyzer. (**F**) Mitochondrial respiration. (**G**) Proton leak respiration. (**H**) ATP-linked respiration. (**I**) Coupling efficiency. Data are represented as mean ± SEM (*n* = 3). (**J**) Intracellular ATP content. Data are represented as mean ± SEM (*n* = 4), and *n*-values represent independent experiments. The statistical significance of mean differences was tested by an unpaired two-tailed Student’s *t*-test to compare two variables. *p* < 0.01 (**), *p* < 0.001 (***), *p* < 0.0001 (****).

**Figure 4 ijms-23-12338-f004:**
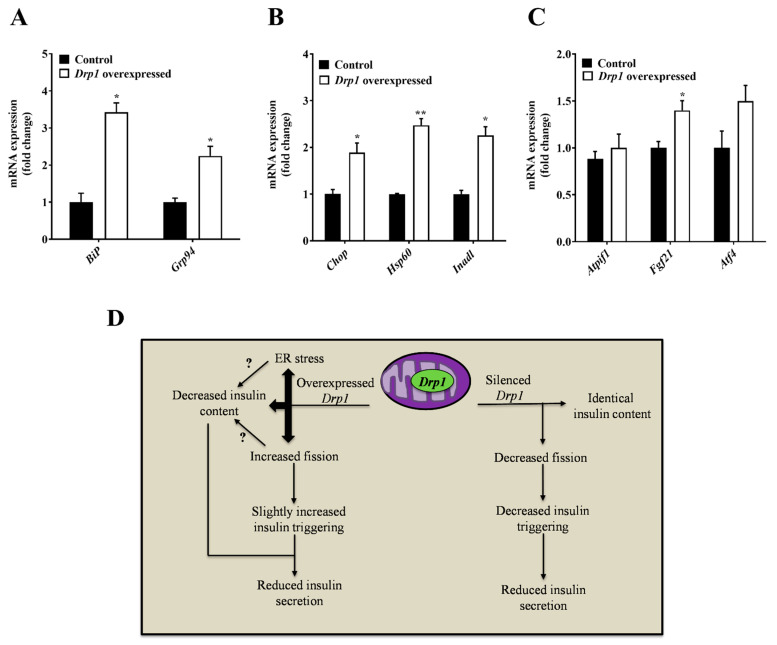
Regulation of mRNA expression of *Drp1*-overexpressed MIN6 cells. MIN6 cells were transiently transfected with the *Drp1* plasmid using the electroporation technique to overexpress *Drp1;* control (black bars) and overexpressed *Drp1* (white bars). (**A**) Relative levels of *BiP* and *Grp94* mRNA were measured by qPCR. (**B**) Relative levels of *Chop, Hsp60,* and *Inadl* mRNA were measured by qPCR. (**C**) Relative levels of *Atpif1, Fgf21,* and *Atf4* mRNA were measured by qPCR. Data are represented as mean ± SEM (*n* = 3), and *n*-values represent independent experiments. (**D**) Schematic model of the impact of *Drp1* in pancreatic *β* cells. The statistical significance of mean differences was tested by an unpaired two-tailed Student’s *t*-test to compare two variables. *p* < 0.05 (*), *p* < 0.01 (**).

## Data Availability

Not applicable.

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
