# Peer review of "Drp1 Overexpression Decreases Insulin Content in Pancreatic MIN6 Cells"

_ijms, 2022, doi:10.3390/ijms232012338_

Round 1

Reviewer 1 Report

The manuscript Drp1 Overexpression Decreases Insulin Content in Pancreatic MIN6 Cells is well validated by results. Drp1 is just association with insulin not directly proven or inhibited or over expressed in plasmid not proving the role.

The Insulin secretion of Drp1 KD MIN6 cells overexpressing Drp1 is not very physiological relevant in vivo model, and the assay is not very sensitive.

The author described that Drp1 overexpression altered mitochondrial morphology but has no effect on mitochondrial respiratory complexes in MIN6 cells, telling that their no direct role just associative role.

Author Response

The manuscript Drp1 Overexpression Decreases Insulin Content in Pancreatic MIN6 Cells is well validated by results. Drp1 is just association with insulin not directly proven or inhibited or over expressed in plasmid not proving the role.

The Insulin secretion of Drp1 KD MIN6 cells overexpressing Drp1 is not very physiological relevant in vivo model, and the assay is not very sensitive.

The author described that Drp1 overexpression altered mitochondrial morphology but has no effect on mitochondrial respiratory complexes in MIN6 cells, telling that their no direct role just associative role.

RESPONSE: We thank the reviewer for his/her appreciation of our ‘well validated’ results. Drp1 KD cells have been used by independent laboratories to assess the impact of Drp1 on insulin secretion (e.g. PMID: 28174288, PMID: 27154223, PMID: 23565276), showing impairment of GSIS. Thus, it is feasible to test whether Drp1 overexpression would improve insulin secretion for translational purposes.

Reviewer 2 Report

The paper from Kabra and Jastroch describes the consequences of DRP1 overexpression on insulin content in the pancreatic cell line MIN6:

  1. The authors show that cell stably KD for DRP1 displays a decreased glucose-stimulated insulin secretion (GSIS). The authors describe that rescue with WT DRP1 and overexpression of DRP1 do decrease the GSIS response at low and high glucose levels.
  2. The authors addressed if the overexpression of DRP1 causes changes in mitochondrial morphology and mitochondrial dynamics genes. As reported by several groups, overexpression of DRP1 caused mitochondrial fragmentation in MIN6 cells without altering the mRNA abundance of the proteins MFN1, MFN2, and OPA1.
  3. The authors found that overexpression reduced insulin content without altering mitochondrial metabolism. For this reason, the authors checked genes related to ER-Stress, since ER-Stress can cause changes in insulin content, and found that DRP1 overexpression induces ER-Stress.
  4. The authors conclude that changes in mitochondrial dynamics could alter the insulin content on MIN6 and beta-pancreatic cells, questioning the potential therapeutic role of DRP1. 

Overall, the authors provide good evidence about DRP1 and the control of insulin content, and probably a non-canonical role for DRP1. However, there are major and minor concerns about the conclusion stated by the authors. These are detailed in the following points

Major:

The data provided is not enough to conclude that the disturbance in mitochondrial dynamics influences insulin synthesis since this can also be explained only by the ER stress caused by DRP1 and not the morphological changes induced in the mitochondria. The fact that there are no metabolic changes in mitochondrial respiration upon DRP1 overexpression reinforces that the ER and not mitochondrial morphology is what matters. 

To test this the role of mitochondria morphology and ER, I suggest: 

  1. Acutely express a DN-DRP1 to test if the observed effects are related to DRP1 activity. It would be needed to check if the DN-DRP1 also causes ER-Stress.
  2. Silence DRP1 acutely to see if there is any effect on insulin secretion/content upon acute change in mitochondrial morphology.
  3. Acutely induce ER stress by using thapsigargin or tunicamycin in the presence of DRP1 overexpression or DN-DRP1. These experiments would provide evidence about the role of DRP1 on insulin content in these cells. 
  4. Control the DRP1 levels in the ER and mitochondria during Overexpression levels

The PKR/elF2a/FGF21 pathway, mentioned in the abstract as one of the findings, is only addressed by measuring one gene by qPCR. To test this, I suggest: 

  1. A complete evaluation of this pathway by qPCR and Western bott is needed. Include positive and negative controls. Also, in the discussion, an explanation of how DRP1 might be involved in this pathway needs to be included. 
  2. It is essential to test that protein overexpression is not causing ER stress per se. I suggest overexpressing a non-related protein as a control.

Minor

- Please include the MW in all the western blots in the figures and original images

- The labeling in figure one sometimes is confusing. Please label the black bar as a WT or Mock because it is not clear from the text or the legend

- Please explain why a t-test was used to analyze figure 1 and not an ANOVA test.

-The DRP1 KD condition shows no apparent rescue on the insulin secretion upon rescue or overexpression with WT DRP1. However, if the fold change is considered, the DRP1 KD + DRP1 starts from a lower basal than the KD cells. Still, they can reach similar levels upon GSIS. Why was this possibility not considered?

- In figure 2, mitochondrial fragmentation is unclear from the images; please use a grey scale. Also, provide an inset showing mitochondrial morphology

- Since the western blot is saturated, it is difficult to see if there are changes in the protein expression since the signal's dynamic range is lost. I suggest including a less saturated blot.

- Why Complex I and II were not detected in the WB?

- In Fig3, please include representative traces of the seahorse experiments 

- In Fig4, please evaluate the pathway proposed by the authors by Western blot and qPCR PKR/elF2a/FGF21

- A scheme summarizing the findings would be convenient 

- Please carefully revise the methods. The catalog number provided for DRP1 belongs to a protein involved in protein synthesis, not the DRP1 involved in mitochondrial dynamics.

- In line 38, there is a typo for metabolism

Round 2

Reviewer 2 Report

I appreciate the authors have addressed most of the minor and major concerns. The work shows great improvement. However, I still have some minor worries

-The PKR/eIF2a/FGF21 pathway is mentioned in the abstract as one of the conclusions. The data presented is insufficient to conclude that this pathway is upregulated since only one of the genes was measured. 

-The data is enough to conclude that DRP1 cause ER-stress and UPR due to the upregulation of BIP, CHOP, and Hsp60. 

-The data is enough to conclude that FGF21 is upregulated, but the authors can only suggest that the complete pathway is upregulated because PKR and eIF2a were not measured. I would suggest modifying the abstract to highlight the findings and suggestions. 

-The presentation of the data in this section could be improved. 

-I must insist on the catalog number for the DRP1 antibody described by the authors. 

The catalog number listed in the paper corresponds to the BD Bioscience antibody DRP1, which according to the company, recognizes a protein called Density Related Protein 1, a protein of 20kDa that has nothing to do with mitochondrial dynamics (Catalog number 611738)

https://www.bdbiosciences.com/en-us/products/reagents/microscopy-imaging-reagents/immunofluorescence-reagents/purified-mouse-anti-drp1.611738

-The BD antibody that recognizes DRP1 involved in mitochondrial dynamics is DLP1 (Catalog number 611113)

https://www.bdbiosciences.com/en-us/products/reagents/microscopy-imaging-reagents/immunofluorescence-reagents/purified-mouse-anti-dlp1.611113

Author Response

First of all, we like to thank this reviewer for a really thorough quality review. We hope that we can address all of the valid concerns in this round.

I appreciate the authors have addressed most of the minor and major concerns. The work shows great improvement. However, I still have some minor worries

-The PKR/eIF2a/FGF21 pathway is mentioned in the abstract as one of the conclusions. The data presented is insufficient to conclude that this pathway is upregulated since only one of the genes was measured. 

RESPONSE: We fully agree that our statement is too strong and have amended the conclusion in the abstract “Coherent with previous studies in Drp1-overexpressing muscle cells, we found upregulation of ER stress-related genes (BIP, CHOP, and Hsp60) possibly impacting insulin production in MIN6 cells.”

-The data is enough to conclude that DRP1 cause ER-stress and UPR due to the upregulation of BIP, CHOP, and Hsp60. 

RESPONSE: We agree.

-The data is enough to conclude that FGF21 is upregulated, but the authors can only suggest that the complete pathway is upregulated because PKR and eIF2a were not measured. I would suggest modifying the abstract to highlight the findings and suggestions. 

RESPONSE: We agree.

-The presentation of the data in this section could be improved. 

RESPONSE: Thank you for the general recommendation.

-I must insist on the catalog number for the DRP1 antibody described by the authors. 

The catalog number listed in the paper corresponds to the BD Bioscience antibody DRP1, which according to the company, recognizes a protein called Density Related Protein 1, a protein of 20kDa that has nothing to do with mitochondrial dynamics (Catalog number 611738)

https://www.bdbiosciences.com/en-us/products/reagents/microscopy-imaging-reagents/immunofluorescence-reagents/purified-mouse-anti-drp1.611738

-The BD antibody that recognizes DRP1 involved in mitochondrial dynamics is DLP1 (Catalog number 611113)

https://www.bdbiosciences.com/en-us/products/reagents/microscopy-imaging-reagents/immunofluorescence-reagents/purified-mouse-anti-dlp1.611113

RESPONSE: The reviewer is absolutely right and has spotted our error. We were previously puzzled as we only checked the catalogue number of our DRP1 construct, not of the antibody. We corrected the catalogue number which now states the correct antibody.